# Extracellular Cysteines Are Critical to Form Functional Cx46 Hemichannels

**DOI:** 10.3390/ijms23137252

**Published:** 2022-06-29

**Authors:** Ainoa Fernández-Olivares, Eduardo Durán-Jara, Daniel A. Verdugo, Mariana C. Fiori, Guillermo A. Altenberg, Jimmy Stehberg, Iván Alfaro, Juan Francisco Calderón, Mauricio A. Retamal

**Affiliations:** 1Programa de Comunicación Celular en Cáncer, Facultad de Medicina Clínica Alemana, Universidad del Desarrollo, Santiago 7780272, Chile; aifernandezo@udd.cl (A.F.-O.); ialfaro@udd.cl (I.A.); 2Centro de Medicina Regenerativa, Facultad de Medicina Clínica Alemana, Universidad del Desarrollo, Santiago 7780272, Chile; eduardoduran@udd.cl; 3Laboratorio de Neurobiología, Instituto de Ciencias Biomédicas, Facultad de Medicina y Facultad de Ciencias de la Vida, Universidad Andres Bello, Santiago 7780272, Chile; da.verdugo.r@gmail.com (D.A.V.); jstehberg@unab.cl (J.S.); 4Department of Cell Physiology and Molecular Biophysics and Center for Membrane Protein Research, Texas Tech University Health Sciences Center, Lubbock, TX 79430-6551, USA; mariana.fiori@ttuhsc.edu (M.C.F.); g.altenberg@ttuhsc.edu (G.A.A.); 5Centro Científico y Tecnológico de Excelencia Ciencia & Vida, Santiago 7690000, Chile; 6Centro de Genética y Genómica, Facultad de Medicina Clínica Alemana, Universidad del Desarrollo, Santiago 7780272, Chile; 7Centro de Fisiología Celular e Integrativa, Facultad de Medicina Clínica Alemana, Universidad del Desarrollo, Santiago 7690000, Chile

**Keywords:** redox sensing, connexins, extracellular loops, channel permeability, post-translational modification

## Abstract

Connexin (Cxs) hemichannels participate in several physiological and pathological processes, but the molecular mechanisms that control their gating remain elusive. We aimed at determining the role of extracellular cysteines (Cys) in the gating and function of Cx46 hemichannels. We studied Cx46 and mutated all of its extracellular Cys to alanine (Ala) (one at a time) and studied the effects of the Cys mutations on Cx46 expression, localization, and hemichannel activity. Wild-type Cx46 and Cys mutants were expressed at comparable levels, with similar cellular localization. However, functional experiments showed that hemichannels formed by the Cys mutants did not open either in response to membrane depolarization or removal of extracellular divalent cations. Molecular-dynamics simulations showed that Cys mutants may show a possible alteration in the electrostatic potential of the hemichannel pore and an altered disposition of important residues that could contribute to the selectivity and voltage dependency in the hemichannels. Replacement of extracellular Cys resulted in “permanently closed hemichannels”, which is congruent with the inhibition of the Cx46 hemichannel by lipid peroxides, through the oxidation of extracellular Cys. These results point to the modification of extracellular Cys as potential targets for the treatment of Cx46-hemichannel associated pathologies, such as cataracts and cancer, and may shed light into the gating mechanisms of other Cx hemichannels.

## 1. Introduction

Connexins (Cxs) are transmembrane proteins that form two different types of channels: hemichannels and gap-junction channels [1]. Hemichannels are hexamers of Cxs with a pore diameter (~10–14 Å) [2,3,4] that allows permeability to ions and molecules of up to ~1.2 kDa [5]. In consequence, hemichannels mediate the release of signaling molecules from cells such as ATP [6], glutamate [7], reduced glutathione (GSH) [8,9], and nicotinamide adenine dinucleotide (NAD^+^) [10] as well as the uptake of glucose [11] and Ca^2+^ [12]. Due to their permeability to second messengers and metabolites, hemichannels participate in relevant physiological functions, such as the formation of short-term memory [13], memory consolidation [14], gliotransmitter release [15], retina function [16], bone-hormone response and healing [17,18], neuronal-network activity [19], vasopressin synthesis and release [20], CO_2_ sensing [21], etc. Despite these roles, it is well known that hemichannels present a low open probability under physiological conditions [22], which is greatly increased under pathological conditions [23,24] including metabolic stress [25], ischemia [26], inflammation [11], deafness [27], X-linked Charcot-Marie-Tooth disease [28], and skin diseases [29]. This enhanced hemichannel activity induces cell malfunctioning, which can lead to cell death [27,30,31,32]. Therefore, knowledge about the molecular mechanisms of hemichannel gating will contribute to our understanding of the roles of hemichannels under physiological conditions and pave the way to the development of the means to reduce their increased opening under pathological conditions.

For several years, hemichannel properties have been studied using depolarization of the cell membrane to trigger their opening [33]. At positive voltages (>+10 mV in average), it is possible to record hemichannel-mediated currents in cells expressing several Cxs isoforms, including Cx26 [34,35], Cx32 [36,37], Cx43 [22,38], Cx45 [39], and Cx46 [40,41]. Additionally, removal of extracellular Ca^2+^ and Mg^2+^ increases hemichannel activity [33] in cells expressing diverse Cxs isoforms [29,37,42,43,44]. Additionally, hemichannel activity is regulated by posttranslational modifications, with phosphorylation being the most studied [45,46,47]. Our laboratory has been interested in hemichannel regulation by gaseous transmitters [25,48,49,50,51], being among the first to demonstrate that Cx43 hemichannels are S-nitrosylated by nitric oxide (NO) in astrocytes under metabolic inhibition [25]. It was latter demonstrated that Cys271 is S-nitrosylated, when Cx43 is forming gap-junction channels between endothelial and smooth-muscle cells [52] as well as when forming hemichannels in cardiomyocytes [53]. S-nitrosylation induced by NO can also occur in Cx46, and the residue Cys212 appears to be important in this process [51]. Another gaseous transmitter that we have studied is carbon monoxide (CO), which seems to act as a Cx43- and Cx46-hemichannel blocker [50]. In the case of Cx46 hemichannels, the inhibition by CO was fully reversed by membrane-impermeable reducing agents, and still occurred when only native extracellular Cys were present [50]. In another study, we demonstrated that one or more Cx46 extracellular-Cys residues are in their reduced configuration (-SH), as they can be modified by the Cys-selective reagent carboxytetramethylrhodamine to make voltage-clamp fluorometry recordings possible [54]. Here, we aimed at determining the role of Cx46 extracellular Cys in hemichannel function and gating, by replacing the extracellular Cys with Ala, one at a time, and studying their expression, localization, and function. All the mutants formed hemichannels at the plasma membrane, but, unexpectedly, none of them opened in response to the positive membrane potentials or the removal of extracellular divalent cations. From these results, we conclude that the extracellular Cys in Cx46 are fundamental for hemichannel function.

## 2. Results

### 2.1. Mutations of Cx46 Extracellular Cys Do Not Alter Their Expression and Cellular Localization

All Cx46-EGFP cDNAs used for transfections, whether wild-type or Cys-mutated, were expressed, allowing their visualization by fluorescence microscopy. We have previously shown that the fusion of EGFP to the C-terminal end of Cx46 does not interfere with hemichannel function [55]. From here and onwards, we will avoid the use of the term EGFP for all Cx46 variants; Cx46-EGFP will be called “wild-type Cx46” or simply “Cx46” and the Cx46-EGFP mutants will be named “Cys mutants”.

Transfection with DNAs corresponding to Cx46 or its Cys mutants in HeLa cells resulted in their expression, as detected by Western blot analysis (Figure 1A, left panel). In all cases, the molecular weight of the immunoreactive bands was consistent with the migration of the corresponding Cx46 fused to EGFP (~75 kDa). Moreover, normalization of Cx46 levels relative to actin revealed that the expression of the Cys mutants was not statistically different from that of Cx46 (Figure 1A, right panel), suggesting that the Cx46 Cys mutants are transcribed and translated properly. We then studied the mutants’ cellular localization, by imaging their EGFP fluorescence. For all the mutants, the EGFP signal was very similar to that observed in the HeLa cells expressing wild-type Cx46 (Figure 1B). EGFP was present in compartments located close to the nucleus, possibly the Golgi apparatus and/or the endoplasmic reticulum (Figure 1B), with faint green fluorescence present in the cytoplasm and at the edges of the plasma membrane, suggesting that Cx46 was transported to the plasma membrane (Figure 1B, white arrow heads). This is consistent with our previous work demonstrating that wild-type Cx46-EGFP forms functional hemichannels in HeLa cells, as shown by dye-uptake experiments [55]. A similar localization pattern was observed for all the Cx46 Cys mutants (Figure 1B). These results suggest that replacement of an extracellular Cys with Ala does not interfere with the normal expression, cellular distribution, and membrane targeting of Cx46.

To assess the possible presence of functional junction channels, a scrape-loading experiment was performed on Cx46-EGFP transfected cells using DAPI, Lucifer yellow, and carboxyfluorescein (CF). After 20 min of exposure to these dyes, no dye transfer between HeLa cells was observed (Appendix A). These results suggest the absence of gap-junction channels between transfected HeLa cells in our preparation, and that the fluorescence observed at the cell membrane is likely to arise only from hemichannels, although the existence of non-functional gap-junction channels cannot be ruled out.

### 2.2. Mutation of Extracellular Cys of Cx46 Eliminates Hemichannel Sensitivity to Removal of Divalent Cations

As mentioned above, hemichannels open in response to several stimuli, including membrane depolarization and removal of extracellular divalent cations, in particular Ca^2+^ [43,55,56]. Therefore, we compared dye uptake in HeLa cells expressing wild-type Cx46 or its Cys mutants under control conditions (normal Ca^2+^ and Mg^2+^ concentrations) vs. that in cells exposed to divalent cation-free solution (DCFS). Under control conditions, the rate of dye uptake in cells expressing wild-type Cx46 was 0.30 ± 0.07 AU/s (Figure 2A, Control), which significantly increased to 1.51 ± 0.21 AU/s in DCFS (Figure 2B, DCFS). Interestingly, exposure to DCFS had no effect on the rate of dye uptake in cells expressing the Cys mutants (Figure 2A,B). These results suggest that the Cys mutants form hemichannels that are not functional or, at least, they are not sensitive to extracellular divalent cations.

As DAPI has a net positive charge (+2) under physiological pH, we also attempted to determine the effects of Cys mutations on the permeability of hemichannels to a probe that displays negative charge. For this we chose CF, which has a net negative charge of −2 at physiological pH. Exposure of HeLa cells transfected with Cx46 WT or Cys mutants showed very low permeability to CF under normal conditions and in DCFS (Appendix A), with very subtle differences in EGFP levels between Cx46 WT, C54A, and C61A. These results suggest that Cx46 hemichannels show a very low permeability to CF. Since all the Cx46 transfected cells showed very little CF uptake, it was not possible to evaluate differences between the Cx46 WT and Cys mutants. This suggests that in our hand, Cx46 hemichannels are not permeable to negatively charged molecules with a MW ≥ 376 g/mol.

### 2.3. Extracellular Cys Mutants of Cx46 Are Targeted to the Plasma Membrane

One possibility that could explain the lack of response of Cx46 Cys mutants to DCFS exposure is that they may not reach the plasma membrane. To address this possibility, we performed an assay using a membrane-impermeable biotinylation reagent that should react only with amino acids accessible from the extracellular solution. As expected, wild-type Cx46 was biotinylated and yielded a Cx46 immunoreactive band of ~75 kDa molecular mass (Figure 3, B lines). All the Cys mutants were biotinylated as well and showed molecular weights similar to that of wild-type Cx46. These results strongly suggest that the Cys mutants are present at the plasma membrane, consistent with the immunofluorescence results in Figure 1B. As a negative control of plasma-membrane biotinylation, we determined the presence of the ER-membrane resident-protein PERK in our samples. PERK was not detected in the biotinylated samples (Figure 3, B lines), suggesting that there was no contamination with cytoplasmatic proteins. Considering the immunofluorescence data and these results, we concluded that the Cx46 Cys mutants are properly targeted to the plasma membrane, where they likely form hemichannels that remain closed in the absence of extracellular divalent cations.

### 2.4. Mutation of Extracellular Cys of Cx46 Eliminates Hemichannel Sensitivity to Plasma-Membrane Depolarization

Xenopus laevis oocytes constitute a classic model for studying hemichannels because they allow for the recording of large currents [56]. Currents were measured in oocytes transfected with cRNA corresponding to wild-type Cx46 or Cys mutants, which were depolarized from −60 to +60 mV. Oocytes expressing wild-type Cx46 showed an evident outward current only at depolarizations higher than +20 mV (Figure 4A). The maximal hemichannel current recorded at +60 mV was 1.83 ± 0.38 mA, *n* = 7 (Figure 4B). In contrast, oocytes injected only with the antisense against the native Cx38 show maximal currents that were 10 times smaller (0.16 ± 0.03 mA, *n* = 8) (Figure 4A,B). Congruent with the dye-uptake results in HeLa cells, oocytes expressing most Cys mutants showed currents very similar to those observed in the control Cx38 -/- oocytes (C65A: 0.12 ± 0.03 mA, *n* = 8; C187A: 0.13 ± 0.02 mA, *n* = 8; C192A: 0.15 ± 0.08 mA, *n* = 8; and C198A; 0.20 ± 0.08 mA, *n* = 7) (Figure 4A,B). In the case of the C59A (0.34 ± 0.10 mA, *n* = 8) and C61A (0.39 ± 0.09 mA, *n* = 8) mutants, their values were higher, but they were still much smaller than those of the wild-type Cx46-expressing oocytes (Figure 4A,B). These data demonstrate that single extracellular Cys mutations inhibit the response of Cx46 hemichannels to membrane depolarization.

### 2.5. In Silico Modeling of Hemichannels Formed by Extracellular Cys Mutants of Cx46 Identify Small Changes in Pore Shape and Electrostatic Potential

To understand the molecular mechanisms by which the substitution of a Cys residue with Ala could inhibit the flow of DAPI through Cx46 hemichannels, we decided to perform a molecular-dynamic (MD) study using human Cx46 as a model. After 50 ns of MD, we observed that the substitution from Cys to Ala induced changes in the vestibule of the pore (Figure 5A), along with minor changes in the minimum pore diameter for almost all mutants (Cx46: 4.85 Å; C54A: 5.02 Å; C61A: 4.01 Å; C65A: 4.53 Å; C181A: 5.11 Å; C186A: 4.00 Å; and C192A: 4.46 Å). However, it is unlikely that such small changes could be the principal cause for the apparent loss of permeability to DAPI, although this possibility cannot be readily ruled out for most mutants. Similarly, Cys mutants displayed modest changes in their pores’ electrostatic potential, compared to the wild-type Cx46 hemichannels (Figure 5B). In particular, C61A and C192A mutants gained electronegativity (shifted to a more intense red), while C181A and C186A lost electronegativity (shifted to a lighter red). The above changes do not appear to be large enough to induce the complete loss of hemichannel permeability, although their contribution cannot be ruled out.

### 2.6. Extracellular Cys Mutants Display Changes in the Parahelix 47–61 Segment In Silico

The parahelix TM1-EL1 segment (part of transmembrane helix 1 and extracellular loop 1) has an important role in the selectivity, conductance, and voltage sensing of several Cxs, including Cx46 [3], and in the calcium sensitivity of Cx26 [57]. Therefore, we decided to study, by means of MD, whether this segment could be altered when Cx46 extracellular Cys are mutated (Figure 6A). As previously described for Cx43 [58], we found that the amino acids at positions 50, 53, and 62 are facing the pore entry (Figure 6A). A zoom of this segment revealed that, to some extent, all the mutants modified the space distribution of the amino acids in segments 47–61 (Figure 6B). These results suggest that Cys mutations could alter the atomic distribution in an important segment, which, as mentioned earlier, participates in the control of hemichannel opening/closing.

## 3. Discussion

In this work we found that the replacement of the extracellular Cys of Cx46 with Ala seems to produce hemichannels that show a dramatic decrease in permeability to DAPI, small inorganic ions in the face of plasma-membrane depolarization, and removal of extracellular divalent cations. This loss of detectable hemichannel activity was not due to a decrease in protein expression or the impairment of transport of the mutants to the plasma membrane. Moreover, MD simulations suggest that these changes, at least for DAPI permeability, could be associated with changes in the 3D disposition of the EC1-TM1 segment (47–63 aa), which contributes to the permeability, voltage, and calcium regulation [59].

In 2004, Bao and co-workers found that hemichannels formed by a Cx43 without extracellular Cys are functional, as measured by the capacity of Xenopus oocytes to uptake carboxyfluorescein from the extracellular media [60]. Contrary to their observations, here we found that Cx46, in which a single extracellular Cys was mutated to Ala, seems to render non-functional hemichannels. We have previously proposed that extracellular Cys carbonylation (i.e., due to a posttranslational modification induced by lipid peroxides) produces a closure of Cx46 hemichannels because of the absence of voltage sensing and a lack of response to the removal of extracellular divalent cations [50,54,55]. Therefore, it seems possible that the oxidation of extracellular Cys thiolates of Cx46 results in hemichannel closure, and that substitution of a single Cys with Ala could result in conformational changes similar to those induced by Cys oxidation, or that the unpaired free remaining Cys could become more prone to oxidization; these are hypotheses that could be tested in future studies.

The apparent closed state of the hemichannels formed by the Cx46 Cys mutants could result from hemichannels that can open but display drastically reduced permeability to DAPI and small inorganic ions, or hemichannels that do not open because they become insensitive to extracellular divalent cations and plasma-membrane depolarization. An alternative explanation is that Cys mutants may interfere with the hemichannel oligomerization and/or trafficking to the plasma membrane. Problems in oligomerization and trafficking have been reported for Cx46 in osteoblastic cells [61]. These two latter possibilities are unlikely as our results clearly show that Cx46 can be biotinylated at the plasma membrane and are observed at the edge of cell membranes.

In the early nineties, Dahl and co-workers demonstrated that disulfide bonds between Cx32 extracellular Cys are important for gap-junction-channel formation [62]. Site-directed mutagenesis showed that Cx32 extracellular Cys residues form three disulfide bonds between extracellular loops 1 (EL1) and 2 (EL2), creating an antiparallel β sheet [63]. Recently, László and colleagues showed through MD analyzes that in Cx43 the extracellular Cys form disulfide bonds which connect EL1 and EL2. Interestingly, these Cys-Cys pairs were shown to interact with amino acids located in the parahelix (i.e 53R, 54C, 55N, and 64V). These interactions formed what they called “stabilization centers”, which are important for gap-junction-channel formation. Notably, when the disulfide bonds were “open”, changes in the torsion angle of 193P were evident and induced movements of amino acids located at positions 55–57, which disorganized the “stabilization centers”. Therefore, changes in Cys at the EL2 can effectively modulate the structure of the EL1/parahelix region [64]. On the other hand, Cx23 only has 4 extracellular Cys and still do form functional hemichannels [65]. Furthermore, replacement of Cys61 and Cys65 for Ala in Cx37 resulted in the expected absence of gap-junction-channel formation, but also hemichannels that remain closed in DCFS [66]. On the other hand, the reduction in the disulfide bonds in the Cx43 hemichannels using dithiothreitol resulted in hemichannels with increased activity [38]. These results and ours strongly support the idea that extracellular Cys control hemichannel closing and opening and suggest that they could be valuable pharmacological targets for modulating Cx hemichannel activity.

## 4. Materials and Methods

### 4.1. Plasmid Engineering

To introduce point mutations in human Cx46 that change Cys residues into Ala residues, site-directed mutagenesis was performed using the Quikchange II XL kit (Agilent Technologies, Santa Clara, CA, USA) in accordance with the instructions of the manufacturer. Separate mutagenesis reactions were performed with six pairs of primers (Table 1), and the desired point mutations were confirmed by sequencing at the core facilities at Pontificia Universidad Católica de Chile (Santiago, Chile) (Figure 7). A similar protocol using the QuikChange Multisite Site-Directed Mutagenesis kit (Agilent Technologies, Santa Clara, CA, USA) was performed to introduce the equivalent Cys to Ala mutations in rat Cx46 (rCx46), obtained from Dr. Lisa Ebihara (University of Chicago), in the plasmid pSP64T-Cx46. We would like to note that Cx46 Cys residues of human and rat EL2 differ in the primary sequence but share the same spatial disposition (Figure 8).

### 4.2. Xenopus laevis Oocytes Preparation

Female frogs were anaesthetized with 1% tricainemethasulphonate (Sigma-Aldrich, St. Louis, MI, USA). A small incision of about 1 cm was made on one side of the abdomen, carefully cutting the skin and muscles until the ovary was visible. Approximately 2 cm^3^ of ovary was removed and placed in Or2 solution (in mM: 82 NaCl, 3 KCl, 1 CaCl2, 1 MgCl_2_, 5 Hepes, pH 7.4), supplemented with collagenase (0.5 mg/mL), at room temperature for ~90 min under constant rotation. The oocytes were then washed 3 times, each with 10 mL of fresh Or2, and placed in 10 mL of Barth s solution (in mM: 88 NaCl, 1 KCl, 2 CaCl_2_, 0.8 MgCl_2_, 10 Hepes, pH 7.4) under constant rotation for 10 min. Finally, the oocytes were transferred to a 90 mm plastic dish with fresh Barth s solution, supplemented with 0.1 mg/mL gentamycin and 20 units/mL of penicillin and streptomycin.

### 4.3. cRNA Preparation and Injection into Xenopus laevis Oocytes

The cRNA was synthesized from the pSP64 containing the cDNA for wild-type rCx46 or its Cys mutants. First, the pSP64 was cut using the SalI restriction enzyme and then the SP6-directed capped kit was used in accordance with the instructions of the manufacturer (mMESSAGE mMACHINE, Ambion, Austin, TX, USA). Oocytes were injected with 12.5 ng of antisense Cx38 oligonucleotide alone (to block synthesis of the endogenous Cx38), or in combination with 50 ng of cRNA coding, for one of the following: wild-type rCx46 or its mutants, C54A, C61A, C65A, C187A, C192A, or C198A. After cRNA injection, the oocytes were maintained in Barth s solution supplemented with 0.1 mg/mL gentamycin and 20 units/mL of penicillin and streptomycin for 24–48 h, before experimental measurements.

### 4.4. Whole-Cell Electrophysiological Recordings

Hemichannel currents were measured at room temperature from oocytes in ND96 solution (in mM: 96 NaCl, 2 KCl, 1.8 CaCl_2_, and 5 Hepes/NaOH, pH 7.4), using pClamp 10/Digidata 1440A A/D Board (Molecular Devices, Foster City, CA, USA) for data acquisition and analysis. Currents were elicited by 15 s square pulses, ranging from −50 mV to +60 mV in 20 mV steps, from a holding voltage of −60 mV, and with 10 s intervals between pulses. Current-voltage (I–V) relationships were calculated from the current values at the end of the pulses.

### 4.5. Transfection of HeLa Cells

HeLa cells were seeded in 6-well plates and 24–48 h later (at ~60% confluence) were transfected with 1 μg of pCMV-Cx46-EGFP (enhanced green fluorescent protein) vector (hCx46, Promega, Madison, WI, USA) or each of the extracellular Cys mutants using Lipofectamine 2000 (ThermoFisher Scientific, Waltham, MA, USA) in accordance with the instructions of the manufacturer, with slight modifications. Briefly, Optimem medium (ThermoFisher Scientific, Waltham, MA, USA), plus Lipofectamine and Optimem medium as well as the plasmid, were mixed for 5 min at room temperature. Then, these two mixtures were put together and incubated for 25 min at room temperature. Finally, the cells  culture medium was replaced with Optimem, and the transfection mixture was added. After 4 h, the cells were incubated in DMEM plus 10% SFB at 37 °C, in 5% CO2, and 48 h later the cells were evaluated for Cx46 expression and function.

### 4.6. Dye Uptake

HeLa cells placed in an epifluorescence inverted microscope (Ti-Eclipse, Nikon Instruments Inc., Melville, NY, USA) adapted for time-lapse studies and were exposed to an external solution containing 10 μM DAPI in recording media. Pictures were taken every 20 s for 20 min. Changes in fluorescence intensity were measured in at least 8 EGFP positive cells per experiment using Nikon NIS-Elements software, and the rate of dye uptake was calculated and plotted using GraphPad 6.0 software. Experiments were performed under control conditions (recording media which contain: 140 mM NaCl, 4 mM KCl, 2 mM CaCl_2_, 1 mM MgCl_2_, 5 mM glucose, and 10 mM Hepes at pH 7.4) or in a nominally divalent cation-free solution (DCFS, which is a recording media without CaCl_2_ and MgCl_2_, plus 5 mM EGTA).

### 4.7. Biotinylation

Three ml of the cell-membrane-impermeable biotinylation reagent sulfo-NHS-SS-biotin (0.5 mg/mL, Thermo Scientific, Waltham, MA, USA) was added to cells in 90 mm dishes after washing three times with ice-cold recording media, pH 8.0. After incubation for 30 min at 4 °C, the cells were washed three times with ice-cold recording media plus 15 mM glycine, pH 7.4, and then they were harvested using RIPA lysis buffer plus protease inhibitors (cOmplete, Mini EDTA-free, Roche Diagnostics, Indianapolis, IN, USA). The lysate was incubated with 50 mL of streptavidin magnetic beads (Roche, Indianapolis, USA) for 1 h at 4 °C, and, after that, 1 mL of washing buffer (recording media, pH 7.2 with 0.1% SDS and 1% Nonidet P-40) was added; the suspension was placed over a magnet, and the supernatant was processed for Western blotting as input control. This washing procedure was repeated three times to remove the resulting supernatant, and then 40 mL of recording media solution plus 0.1 M glycine, pH 2.8, was added to release the bound proteins from the streptavidin beads. The eluted material was vortexed, the mixture was placed over a magnet, the unbound solution was placed in a new Eppendorf tube (1.5 mL), and the pH was adjusted immediately to 7.5 with 10 mL of 1 M Tris. Relative levels of Cx46 present in each sample were assessed by Western blot analysis.

### 4.8. Western Blot Analysis

Cells were lysed using RIPA buffer supplemented with protease inhibitors (Roche, Indianapolis, IN, USA) and sonicated on ice. Protein concentration was determined using a protein-assay kit (ThermoFisher Scientific, Waltham, MA, USA) and read with a Qubit 3.0 Fluorometer (ThermoFisher Scientific, Waltham, MA, USA). Approximately 20 μg of total protein were resolved in a 10% SDS PAGE gel and transferred to a nitrocellulose membrane (Bio-Rad, Hercules, CA, USA). The membranes were incubated overnight with primary antibodies: anti-Cx46 (Santa Cruz Biotechnology, Dallas, TX, USA #C3), anti-PERK (Cell Signaling, Danvers, MA, USA, #C33E10), or anti-actin (Sigma-Aldrich, St. Louis, MI, USA, #A5441). After the primary antibodies were washed with PBS, the membranes were incubated for 1 h at room temperature with the secondary antibodies conjugated to horseradish peroxidase (HRP) (Abcam, Cambridge, UK), and immunoreactivity was visualized using a LI-COR C-Digit chemiluminescence Western blot scanner system (LI-COR, Inc., Lincoln, NE, USA).

### 4.9. EGFP Cellular Localization

As mentioned above, wild-type Cx46 and its Cys mutants were fused to EGFP to facilitate detection and assess cellular localization by fluorescence imaging. To this end, HeLa cells expressing wild-type Cx46 or its mutants were grown in 1.2 mm glass coverslips to a 60–70% confluence. After washing the coverslips three times with PBS, the cells were fixed with 4% paraformaldehyde for 30 min at room temperature. Then, the cells were permeabilized by exposure to PBS plus 1% Trion X-100 for 10 min at room temperature. Finally, the cells were washed three times with PBS and mounted using fluoromount G plus DAPI (to stain cell nuclei) and visualized on a confocal microscope using Fv1200 Olympus Imaging System at Centro Ciencia & Vida (Santiago, Chile).

### 4.10. Molecular Dynamics

The structural model of Cx46 was generated from the amino acid sequence available in the UniPROT database (UNIPROT ID Q9Y6H8). A model for the structure of the Cx46 hemichannel was constructed using the SWISSMODEL tool (https://swissmodel.expasy.org, accessed on 9 May 2022), taking as a template the 7JKC of a Cx46 gap-junction-channel crystalized structure. The C-terminal domain was not included as it has not been fully resolved in the crystal structure. One hemichannel was removed and the remaining hemichannel was loaded into the VMD 1.9.3 software. Subsequently, six different mutants, C54A, C61A, C65A, C181A, C186A, and C192A, were generated by replacing each Cys for Ala. Hence, seven different systems were generated for each Cx46 structure, including the wild-type Cx46 and its 6 Cys mutants. These were embedded in a lipid membrane and submerged in a cubic box with explicit TIP3P water molecules, adding Na^+^ and Cl^−^ ions for neutralization. Given that Cys are structurally critical by producing disulfide bonds, the bonds were kept using a patch between the C54-C192, C61-C186 and C65-C181, except for the disulfide bond formed by the mutated Cys for each mutant. All molecular dynamics simulations were performed using NAMD 2.9 and CharmM36 force field. A 12-Å cut was used for non-linked short-range interactions. The temperature and pressure were maintained at 310 K and 101.3 kPa, using Langevin’s dynamics and the Langevin piston method, respectively. Each time step in the simulation was 1 fs. For each system, an energetic minimization was performed first with the restricted protein for 200 ps and then with the free system for 200 ps. Subsequently, the corresponding molecular dynamics simulations were performed for 50 ns using a HPC Cluster. Systems were prepared for the simulations of Cx46 and all mutants using a similar procedure. Subsequently, the diameter of the pore was calculated with HOLE software (http://www.holeprogram.org, accessed on 9 May 2022), and finally, the electrostatic potential was calculated using the free academic version of the Maestro software 13.1 2021-1, using the Poisson–Boltzmann ESP panel.

### 4.11. Statistical Analysis

Results are expressed as means ± SEM and *n* refers to the number of independent experiments. For statistical analyses, each treatment was compared to its respective control, and significance was determined using a one-way ANOVA or paired Student’s *t*-test, as appropriate. Differences were considered significant when *p* < 0.05.

## Figures and Tables

**Figure 1 ijms-23-07252-f001:**
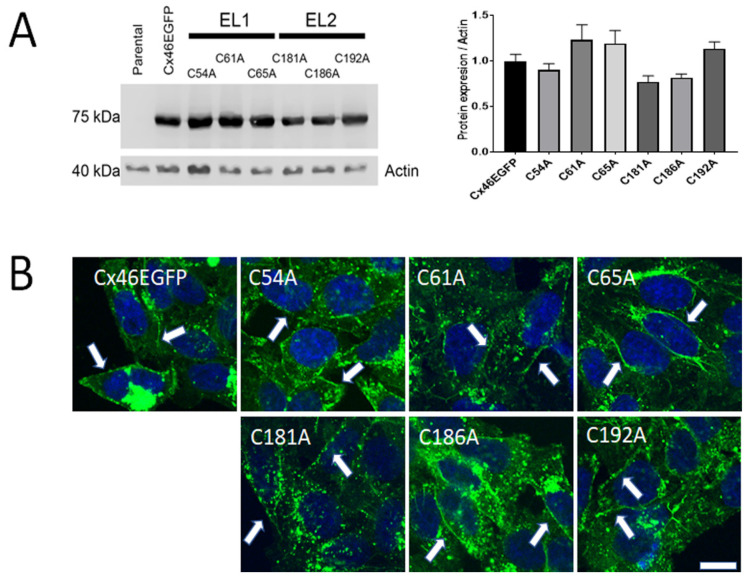
Expression and cellular localization of Cx46 Cys mutants. HeLa cells were transfected with a wild-type or Cys–Ala-mutated human Cx46 DNA fused to EGFP. (**A**) Representative blot of three independent Western blots. Fifty μg of protein were loaded in each line. A ~75-kDa band was detected in all the cases, except for non-transfected HeLa cells (Parental). Actin was used as loading control. The graph on the right presents the densitometric analysis of the three Western blots and shows no statistical differences between protein levels, when comparing Cys mutants and wild-type Cx46 (*n* = 3, n.s = *p* > 0.05). (**B**) Representative images of EGFP fluorescence in HeLa cells transfected with wild-type Cx46 or Cys mutants; nuclei were visualized with DAPI. White arrow heads denote fluorescence at cell-to-cell contacts. Calibration bar = 10 µm.

**Figure 2 ijms-23-07252-f002:**
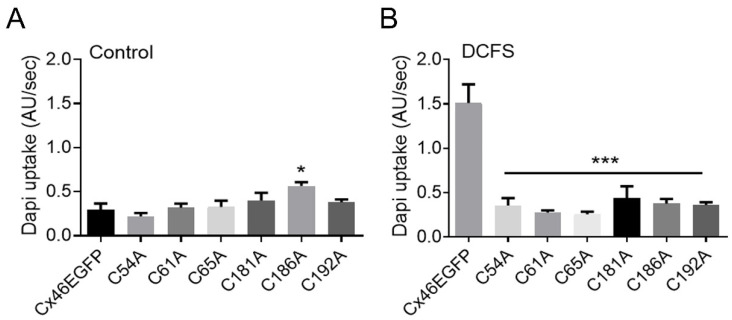
Hemichannels formed by Cys mutants do not open upon exposure to divalent cation free solution (DCFS). HeLa cells transfected with wild-type Cx46 or Cys mutants were placed in recording media containing 10 μM DAPI under control conditions or in DCFS. Pictures of 16 cells were taken every 20 s for 20 min, with 4 replicates. DAPI uptake was calculated as arbitrary units per second (AU/Sec). (**A**) Control conditions. The rate of DAPI uptake was similar in cells expressing all Cys mutants, except for the C186A mutant, which showed a small, but significant higher, DAPI uptake compared to cells expressing wild-type Cx46 (*, *p* < 0.05). (**B**) DCFS. HeLa cells expressing wild-type Cx46 showed an enhanced rate of DAPI uptake, which was not observed in any of the Cys mutants (***, *p* < 0.001) (*n* = 4 for each condition).

**Figure 3 ijms-23-07252-f003:**
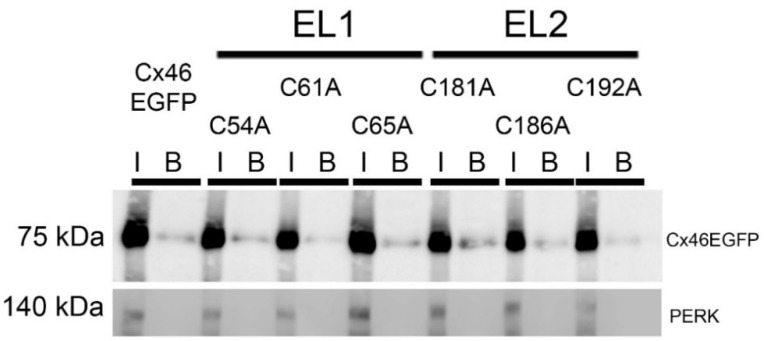
Cys mutants are present at the plasma membrane. Cell-surface biotinylation was performed in cultures of HeLa cells transfected with wild-type Cx46 or Cys mutants, and samples were then analyzed by Western blotting. I (Input): 50 μg of protein before the separation with streptavidin-magnetic beads; B: the whole sample after the separation with streptavidin-magnetic beads. The presence of the cytoplasmatic protein PERK was used as negative control and, as expected, was found only before separation and not in the biotinylated sample. (*n* = 3 independent experiments).

**Figure 4 ijms-23-07252-f004:**
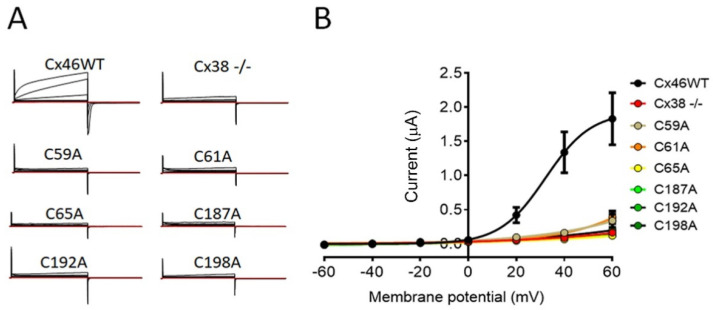
Hemichannels formed by Cys mutants are insensitive to membrane depolarization. Xenopus oocytes were injected with an oligo against the endogenous Cx38 alone (Cx38 -/-) or in combination with cRNA of wild-type Cx46 or Cys mutants. Hemichannel currents were recorded using dual-electrode voltage clamp after 24 h in 8 oocytes. (**A**) Representative recordings showing that only hemichannels formed of wild-type Cx46 open after membrane depolarization. (**B**) I/V plot summarizing results of the 8 independent experiments. The Boltzmann equation was used to fit current values.

**Figure 5 ijms-23-07252-f005:**
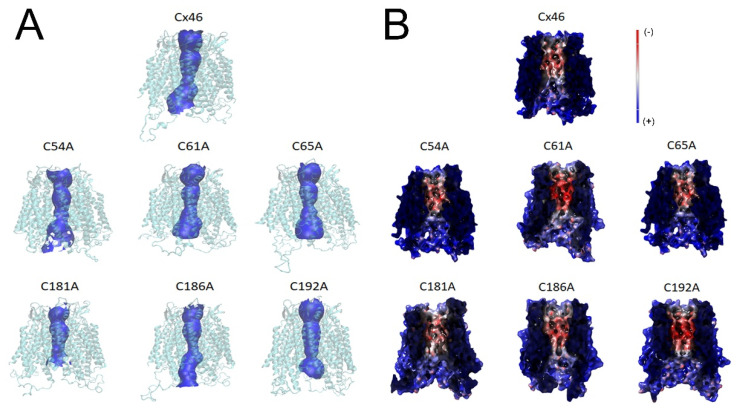
MD simulations suggest discreet changes in both pore size and electrostatic potential in hemichannels formed by Cys mutants. Human Cx46 50 ns MD models. (**A**) Hemichannel tertiary structure is shown in light blue, while the pore profile is shown in blue. (**B**) Electrostatic potential of the hemichannel pore with negative and positive potentials represented in red and blue colors, respectively. Note that small red changes are observed in all the mutants, which were more evident in C61A and C192A.

**Figure 6 ijms-23-07252-f006:**
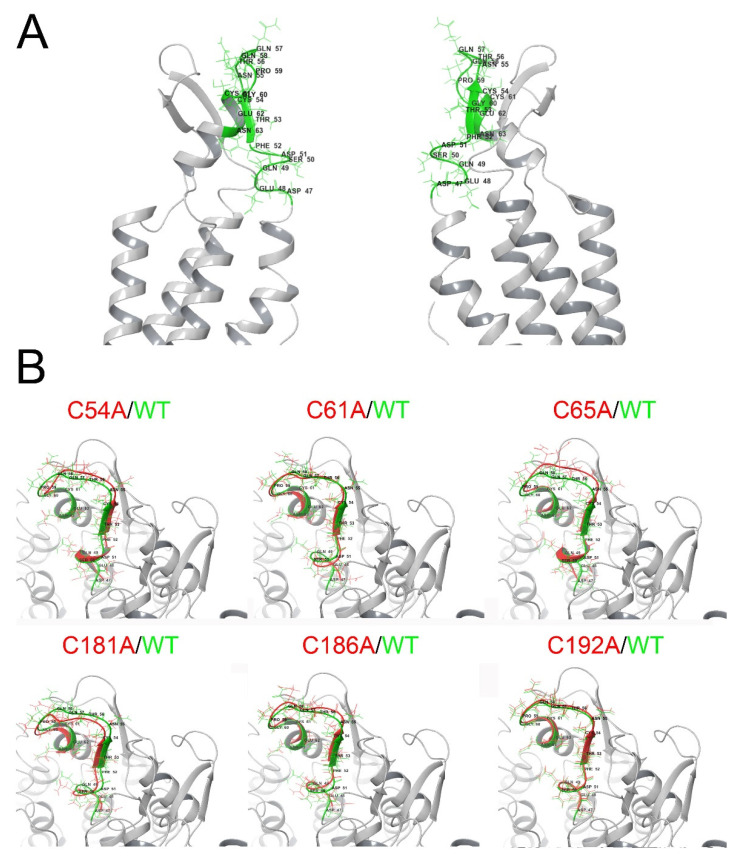
MD simulations suggest changes in the 3D disposition of the TM1-EL1 parahelix segment. Human Cx46 50-ns MD models. (**A**) Representation of a Cx46 hemichannel showing the location of the parahelix (green segment). (**B**) Zoom of the parahelix segment (the segment between amino acids 47 and 61 was analyzed). The configuration of this segment in the wild-type Cx46 (green) is superimposed on the parahelix model for each Cys mutant (red).

**Figure 7 ijms-23-07252-f007:**
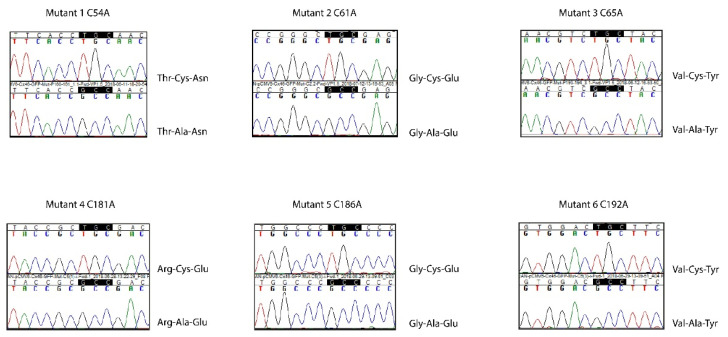
Sequencing results from Cx46 Cys mutants. cDNA plasmids containing the mutagenesis products were sequenced. For each panel, the upper electropherogram shows the sequence for human wild-type Cx46, and the bottom part shows the sequence obtained for the mutants.

**Figure 8 ijms-23-07252-f008:**
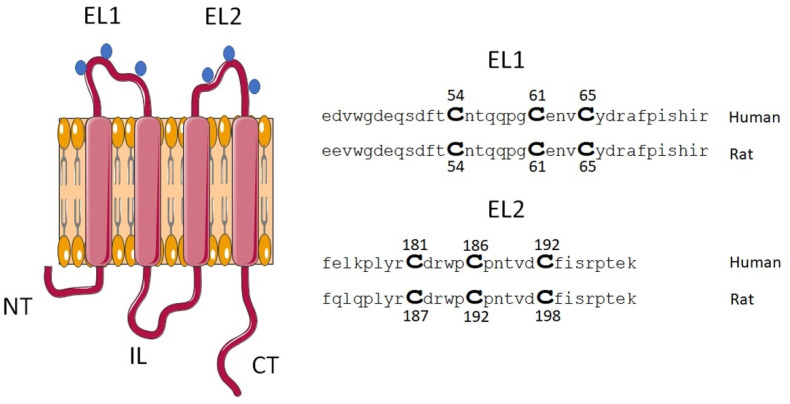
Schematic representation of the localization of extracellular Cys of Cx46. The diagram shows the topological disposition of Cx46 at the plasma membrane and the approximate location of extracellular Cys residues (blue dots). In this study, we used Cx46 derived from human and rat, where Cys located at the EL1 are identical, however, the Cys at EL2 show differences in location within the primary sequence, which are homologous in function.

**Table 1 ijms-23-07252-t001:** Site-directed mutagenesis primers for hCx46 extracellular Cys mutations.

Primer Name	5′ to 3′ Sequence
Cx46-C54A Forward	cagtcagacttcaccgccaacacccagcagcc
Cx46-C54A Reverse	ggctgctgggtgttggcggtgaagtctgactg
Cx46-C61A Forward	ccagcagccgggcgccgagaacgtctgc
Cx46-C61A Reverse	gcagacgttctcggcgcccggctgctgg
Cx46-C65A Forward	gctgcgagaacgtcgcctacgacagggcct
Cx46-C65A Reverse	aggccctgtcgtaggcgacgttctcgcagc
Cx46-C181A Forward	ccgctctaccgcgccgaccgctggcc
Cx46-C181A Reverse	ggccagcggtcggcgcggtagagcgg
Cx46-C186A Forward	cgaccgctggcccgcccccaacacggtg
Cx46-C186A Reverse	caccgtgttgggggcgggccagcggtcg
Cx46-C192A Forward	cccaacacggtggacgccttcatctccaggcc
Cx46-C192A Reverse	ggcctggagatgaaggcgtccaccgtgttggg

## Data Availability

All data will be available as soon as requested.

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
