# Peer review of "Extracellular Cysteines Are Critical to Form Functional Cx46 Hemichannels"

_ijms, 2022, doi:10.3390/ijms23137252_

Round 1
Reviewer 1 Report
In the manuscript by Fernández-Olivares et al the authors aim to understand the physiological control of connexin (Cx46) hemichannels in model systems of channel function. Connexin hemichannels are reported to be important signaling moieties in the membranes of cells, allowing for the exchange of molecules including ATP, glutamate, and others. In their study, the authors created individual mutations at each of the extracellular cysteines, known to be involved in the structural organization of the channel. These are also known, in other connexins, to be required for head-to-head docking of hemichannels to form gap junctions. The data demonstrate, Cx46 Cys-mutations reach the plasma membrane in cell culture and can be expressed in oocytes. Despite equal expression to WT-Cx46, the mutated isoforms show no evidence of hemichannel activity as demonstrated by a loss of DAPI uptake in the presence of Ca2+/Mg2+ free media, as well as a loss of channel current measured in Oocytes. Molecular dynamic modeling data reveal minor changes in pore size and electrostatic potentials following cys mutation. Molecular dynamics suggests that mutations alter the spatial distribution of amino acids 47-61 in TM1 which may be involved in molecule selectivity, conductance, and voltage sensing. In general, the manuscript is well written. The data demonstrate a loss of channel functions following extracellular cysteine mutation and the conclusions support the idea that extracellular cysteines are required for Cx46 hemichannel function.
Specific comments:
1. Data appear to demonstrate the formation of aggregates between cells in HeLa cells, similar to what is observed for gap junctions. Does a loss of a single cysteine completely inhibit Cx46 gap junction channel formation and signaling?
2. Mutations of Cx46-Cys cause complete loss of hemichannel function (DAPI and small ions), which differs from previous studies in Cx43 to carboxyfluorescein (Bao 2004). It is not clear why differences exist between the models. Given that structural models predict altered molecule selectivity and conductance, were other molecules tested? DAPI has a net positive charge (+2) would differences be expected for channel-permeant molecules such as Lucifer Yellow (-2)?
3. Figure 1 has an apparent error. It is not clear what the difference between B and C are? 1C is not described in the figure legend or text. Scale bars are not visible on the images.
Reviewer 2 Report
This manuscript provides interesting results, although some key points have to be addressed (please find comments attached)
Comments for author:
Title: Extracellular cysteines are critical to form functional Cx46 hemi-
channels
p.1 Abstract – Methods: We studied Cx46 and mutated each of its extracellular Cys to alanine (Ala)
(one at a time)
p.1 Abstract – Results: Molecular dynamics simulations showed that Cys-mutants
p.2 comma missing between Cx43[22,38], Cx45[39]
p.2 removal of extracellular Ca2+ and Mg2+ increases hemichannel activity
p.2 Additionally, hemichannel activity is regulated by posttranslational modifications, with phosphorylation being the most studied
p.2 We have previously shown that the fusion of EGFP to the C-terminal end of Cx46 does not to interfere with hemichannel function[55]
p.2 From here and onwards,
Legend to Figure 1: A ~75 kDa band was detected in
The graph on the right shows corresponds to the densitometric analysis
Legend to Fig. 1C is missing
p.3 normalization of Cx46 levels relatives to actin
p.3 Fig 1B, white arrow heads: shown in Fig. 1C, not B
Further comments/questions regarding Fig.2:
Dapi carries a divalent positive charge, were other types of dyes (uncharged or negatively charged) also tested? Maybe the Cys to Ala mutation affects charge selectivity of Cx46.
p.5 2.4 Mutation of extracellular Cys of Cx46 eliminates hemichannel sensitivity to plasma membrane depolarization
C54A (0.34 ± 0.10 mA, n=8) C54A is not shown in Fig.4, is this a typo and C59A is meant here?
Fig.4 Current A should be written as Current (A) in accordance with x-axis labeling
p.6 and Fig.5 In particular, C61A and C192A mutants were more electronegative (shifted to a more intense red), while C181A and C186A were more positive (shifted to a lighter red).
This is in contradiction with:
B) Electrostatic potential of the hemichannel pore with positive and negative potentials represented in red and blue
here positive = red, negative = blue
According to this color code the above sentence should read: In particular, C61A and C192A mutants were less electronegative (shifted to a more intense red), while C181A and C186A were less positive (shifted to a lighter red).
Legend to Fig.8:
In this study, we used Cx46 derived from human and rat, where Cys located at the EL1 are identical, but the Cys at EL2 show differences in location within the primary sequence, but they are homologous in function.
suggested rewording:
In this study, we used Cx46 derived from human and rat, where Cys located at the EL1 are identical, but the Cys at EL2 show differences in location within the primary sequence, which are homologous in function.
Fig.6 Based on the crystal structure of Cx26, residues of TM1-E1 boundary and N-terminal half of E1 form the inner wall of the extracellular entrance of the pore. The authors need to explain in further detail how the Cys mutants in residues 181, 186, 192 (which are part of EL2) can lead to changes in the parahelix TM1-EL1 segment. Maybe additional figures could be helpful? The space distribution of the amino acids in the segment 47-61 shown in Fig 6B appears to be only slightly modified between Cx46 WT and the various mutants.
The reviewer would like to draw the authors’ attention to the ion channel redox model by BS Marinov (PMID: 1709974). Here, switching thiols from an oxidized to a reduced state (or vice versa) is supposed to change protein conformation and ion channel gating. For maxi-K channels where the activity is highly dependent on Ca2+ , it was hypothesized that SH groups close to the Ca2+-binding region of the channel might contribute to the Ca2+ binding affinity depeding on their ionization state (PMID: 9234169). Did the authors take into account that the Ca2+ sensitivity of the Cx46 mutants might be affected? Nominally divalent cation-free solution (DCFS) might still contain M of contaminant Ca2+ . Did the authors check if an EGTA-buffered low Ca2+ extracellular solution was able to open hemichannels formed by Cys mutants in dye uptake experiments?
References:
Ref. 16 remove (80-.) after Science
Ref. 38 Journal name missing
Ref. 61-63 doi missing
Round 2
Reviewer 2 Report
Please find my comments attached
Comments for author:
The authors did a great job improving the manuscript; I just have a few remaining suggestions. The authors should adapt the manuscript accordingly before finalization and publication.
l.6-21 Check affiliations for consistency, sometimes Santiago is separated by commas, sometimes by dots
l.28 We studied Cx46 and mutated each of its extracellular Cys
l.100 the molecular weight of the immunoreactive bands were was consistent
l.129/30 The graph on the right shows presents the densitometric analysis of the three Western blots and shows no statistical differences between protein levels
l.196/7 Currents were measured in oocytes transfected with cRNA corresponding to wild-type Cx46 or Cys mutants which were depolarized
l.233 while C181A and C186A loss lose electronegativity
l.344-6 In this study, we used Cx46 derived from human and rat, where Cys located at the EL1 are identical, but however the Cys at EL2 show differences in location within the primary sequence, but which are homologous in function.
l.502 ref 7 is missing DOI
l.523 ref 15 is missing DOI
l.562 ref 33 is missing DOI
l.564 ref 34 is missing DOI
l.590 ref 47 incomplete
Author Response
We would like to thank your dedication and thoroughness , to make this work much better than originally presented. Thanks a lot.
Changes in the new version are in yellow
